# A Review on UAS Trajectory Estimation Using Decentralized Multi-Sensor Systems Based on Robotic Total Stations

**DOI:** 10.3390/s25133838

**Published:** 2025-06-20

**Authors:** Lucas Dammert, Tomas Thalmann, David Monetti, Hans-Berndt Neuner, Gottfried Mandlburger

**Affiliations:** 1Research Units Engineering Geodesy and Photogrammetry, Department Geodesy and Geoinformation, TU Wien, 1040 Vienna, Austria; tomas.thalmann@geo.tuwien.ac.at (T.T.); hans.neuner@geo.tuwien.ac.at (H.-B.N.); gottfried.mandlburger@geo.tuwien.ac.at (G.M.); 2Skyability GmbH, 7011 Siegendorf, Austria; david.monetti@skyability.com

**Keywords:** 6-DoF trajectory estimation, image-assisted total station, sensor synchronization, UAV

## Abstract

In our contribution, we conduct a thematic literature review on trajectory estimation using a decentralized multi-sensor system based on robotic total stations (RTS) with a focus on unmanned aerial system (UAS) platforms. While RTS are commonly used for trajectory estimation in areas where GNSS is not sufficiently accurate or is unavailable, they are rarely used for UAS trajectory estimation. Extending the RTS with integrated camera images allows for UAS pose estimation (position and orientation). We review existing research on the entire RTS measurement processes, including time synchronization, atmospheric refraction, prism interaction, and RTS-based image evaluation. Additionally, we focus on integrated trajectory estimation using UAS onboard measurements such as IMU and laser scanning data. Although many existing articles address individual steps of the decentralized multi-sensor system, we demonstrate that a combination of existing works related to UAS trajectory estimation and RTS calibration is needed to allow for trajectory estimation at sub-cm and sub-0.01 gon accuracies, and we identify the challenges that must be addressed. Investigations into the use of RTS for kinematic tasks must be extended to realistic distances (approx. 300–500 m) and speeds (>2.5 m s^−1^). In particular, image acquisition with the integrated camera must be extended by a time synchronization approach. As to the estimation of UAS orientation based on RTS camera images, the results of initial simulation studies must be validated by field tests, and existing approaches for integrated trajectory estimation must be adapted to optimally integrate RTS data.

## 1. Introduction

State-of-the-art mapping systems (e.g., laser scanner or cameras) achieve accuracies in their measurements that outperform the geo-referencing accuracy of conventional GNSS-based multi-sensor systems [1,2,3]. To leverage the potential of these mapping systems, more accurate geo-referencing is required. One possibility to achieve this is the application of robotic total stations (RTS) for trajectory estimation. The application of RTS as part of a multi-sensor system for trajectory estimation allows for an accurate approach to geo-referencing unmanned aerial system (UAS) data. While this is suitable for use cases where the highest accuracy is required, it can also be used if no precise GNSS signal is available, e.g., for investigating infrastructure such as bridges from below or for water dams. In general, the choice of survey-grade positioning sensors is limited. In contrast to the direct techniques GNSS and RTS, indirect techniques, e.g., simultaneous localization and mapping (SLAM) or bundle block adjustment, require additional measurements in the scene to be geo-referenced. Other (relative) techniques like inertial navigation or visual odometry can only estimate position and orientation changes and lack a superior spatial reference. RTS can provide very high accuracies and flexibility with regard to satellite signal occlusion or jamming. However, they first determine the target position in a local coordinate system, which can then be transformed into a global coordinate frame. RTS rely on line-of-sight towards the target and have a generally limited measurement range of usually about 1000 m, depending on the reflector and RTS model. With our proposed decentralized multi-sensor system, which is based on RTS, we aim to achieve position accuracies at a sub-cm level and orientation accuracies of the UAS of less than 0.01 gon. This review article provides a comprehensive overview of existing research and challenges relevant to trajectory estimation for kinematic platforms using RTS and image-assisted total stations (IATS), respectively, with a focus on estimating the position and orientation of an UAS. In this work, we will use the term IATS to describe RTS that are extended with a camera functionality for photogrammetric measurements.

By integrating data from navigation sensors, such as IMUs, and mapping sensors (e.g., cameras or laser scanners) mounted on a UAS with trajectory data from RTS or IATS, a decentralized multi-sensor system is made up.

Therein, the IATS forms the ground segment, while the UAS and its sensors onboard constitute the kinematic segment of the multi-sensor system. This concept is illustrated in Figure 1, where the ground segment is extended by image observations of the IATS that allow the observation of the UAS orientation.

The spatial separation between the ground and the kinematic segments makes time synchronization between both segments challenging. Even more so, since the high speed of a UAS of up to 10 m s^−1^ requires a particularly precise synchronization, at the level of less than 1 ms, of all segments. Centralized multi-sensor systems combine all sensors directly on a platform and typically realize time synchronization through wired connections, whereas decentralized systems consist of spatially distributed sensors and, thus, require a wireless approach for time synchronization, which can be realized using the satellite-based pulse per second (PPS) signal [4].

The most relevant sources in the context of this decentralized multi-sensor system are given in Table 1 together with their main contributions. Each publication group is identified by a letter, which is also shown in Figure 1, where each publication group is connected to a specific step of the system concept. While each publication group has its own contribution, only the combined consideration of these research results builds the foundation for our decentralized multi-sensor system. However, every one of these publications requires adaptation to be employed in our specific use case.

**Table 1 sensors-25-03838-t001:** Key publications in the context of UAS trajectory estimation with decentralized multi-sensor systems based on robotic total stations. The grouping letters connect the table to the different steps, which are sketched in Figure 1.

Group	Publications	Main Contribution to a Decentralized Multi-Sensor System
A	Thalmann and Neuner [5], Kälin et al. [6], Grimm and Hornung [7], Stempfhuber and Sukale [8]	Time synchronization and latency estimation of RTS and uncertainty assessment of RTS for kinematic measurement scenarios
B	Brocks [9], Kukuvec [10], Hirt et al. [11]	Investigation of uncertainties introduced by atmospheric refraction on RTS measurements
C	Hauth et al. [12], Ehrhart [13], Wagner et al. [14]	Investigation of the capabilities of IATS and image evaluation in combination with RTS measurements
D	Niemeyer et al. [15]	Functional model and simulation regarding the orientation estimation of a UAS based on image observations from IATS
E	Thalmann and Neuner [16]	Robust Kalman Filter for fusion of RTS and IMU data
F	Skaloud and Lichti [2], Brun et al. [17], Pöppl et al. [18]	Holistic trajectory estimation framework that uses correspondences derived from mapping sensors, e.g., laser scanning, to optimize the trajectory

In Group A, the time synchronization of RTS is thoroughly investigated, accompanied by uncertainty investigations of the exhibited latencies, the time synchronization, and the kinematic measurement performance of different RTS. However, the research is limited in terms of range and velocity. Also, it is conducted, with the exception of Kälin et al. [6], in laboratory environments. These publications provide a detailed foundation for the time-synchronized trajectory estimation of moving targets with RTS. Group B focuses on one of the major uncertainty sources for outdoor measurements, namely the atmospheric refraction. The studies address the general impact of refraction and provide a strategy to quantify the exhibited uncertainty. The studies are mainly focused on terrestrial surveying but highlight the importance of an adequate strategy to estimate the refraction effects when conducting high-accuracy measurements. The most important contributions for the orientation estimation of a UAS based on IATS are given by Groups C and D. While Group C describes the general model of IATS together with possible applications and general uncertainty assessments, Group D simulates the derivation of UAS orientations based on IATS image data. In Group C prototypical IATS are used as well as the built-in coaxial camera of modern IATS [13]. Finally, Groups E and F show two different strategies for sensor fusion of RTS with IMU (E) or of RTS, IMU, and LiDAR (F). While the concept of E is real-time capable and directly uses the polar elements observed by the RTS, they neglect the additional information that can be provided by mapping data, such as LiDAR. The publications of Group F makes use of LiDAR data but are not real-time capable.

The following sections review state-of-the-art knowledge regarding key aspects of this decentralized multi-sensor system, with a focus on their impact on trajectory estimation. Section 2 explores measurement influences on RTS, such as atmospheric interactions, prism effects, and the instrumental characteristics of RTS. Additionally, we focus on existing work using RTS to measure kinematic platforms, in particular UAS. Section 3 addresses the critical issue of time synchronization, regarding both the synchronization between the ground segment and the kinematic segment, as well as the temporal synchronization between different subsystems of the RTS itself, such as the time delay between the angle measurement unit and the distance measurement unit [5]. In Section 4 techniques for using image observations of IATS for three-dimensional measurements [19] are discussed as well as observations of platform orientation using images [15]. Section 5 provides a detailed analysis of techniques for integrated trajectory estimation, leveraging observations from cameras or laser scanners onboard the UAS for the trajectory estimation [3,17,18,20]. Finally, the adaptation of the state-of-the-art knowledge for UAS trajectory estimation and the challenges existing for our decentralized multi-sensor system are discussed in Section 6.

## 2. Measurement Process of RTS

An RTS measurement involves a multi-step process that includes several sensor readings and interactions. For example, a distance measurement depends primarily on the electronic distance measurement (EDM) unit, where the emitted laser signal interacts with both the atmosphere and the target prism. Similarly, the automated detection of the reflector, e.g., Leica Automated Target Recognition (ATR and ATRplus) [21] or Trimble Autolock [22], is part of the angle measurement and experiences an influence from atmospheric and prism interaction, too. Understanding and minimizing the effect of these impact sources is crucial for accurate RTS measurements. This section deals first with research about atmospheric effects on the RTS observations followed by the interaction with the 360° prism. Afterward, state-of-the-art research on kinematic measurements with RTS is presented.

### 2.1. Atmospheric Effects

One of the major influences on RTS measurements is the modification of the emitted laser signal as it passes through the atmosphere, commonly referred to as geodetic refraction. This term encompasses all atmospheric effects on the signal, including deviations in the measured distance due to different propagation velocities in the atmosphere and alterations in its direction due to deflection effects [10,23]. The impact of refraction on the measurement is generally proportional to the measured distance.

A study by Hirt et al. [11] investigates the variation of the refraction coefficient over different time spans. The refraction coefficient is a unitless value to quantify terrestrial refraction. It is commonly used in geodesy and can be defined as the radius of a circular arc which is used to simplify the complex curved path of light [9]. They demonstrate that, at a height of approximately 1.8 m above the ground, the coefficient varies with amplitudes of 1–1.5 within 30 min and ranges between −4 and +16 over several hours on sunny days. Under cloud cover, the amplitude decreases to 0.5 over 30 min and ranges only from −2 to +5 over a span of several hours. They state that applying the commonly used refraction coefficient of 0.13 is insufficient when performing accurate measurements. However, these results are particularly valid for observation heights of 1.8 m, which are subject to significant diurnal variations in surface temperature [9]. In higher and intermediate atmospheric layers (above 20 m), variations in the refractive coefficient are expected to be less pronounced. Thus, while the findings of Hirt et al. [11] provide valuable insights into refraction effects at low altitudes, they are not directly applicable to trajectory measurements of a UAS, where measurement heights may reach up to 120 m. Nonetheless, the study underscores the substantial impact refractive effects can have on long-range measurements with geodetic total stations. In Möser et al. [24], a correction formula (Equation (Equation 1)) is provided for near-horizontal sights. This simplified correction formula for vertical angle refraction $k_{R}$ is based on the refraction coefficient *k*, the measured distance *s*, and the radius of the Earth *R* (≈$6.379\times 10^{6}$ m):(1)$$k_{R}=\frac{k\;\cdot \;s^{2}}{2R}$$
with(2)$$k=503\;\cdot \;\frac{p}{T^{2}}\;\cdot \;\left(0.0342+\frac{dT}{dh}\right)$$
where *p* is the air pressure in [hPa], *T* the temperature in [K], and dT/dh the vertical temperature gradient.

As shown in Equation (Equation 1), a fluctuating refraction coefficient with amplitudes higher than 0.5 can significantly affect measurements over longer distances, showing the insufficiency of a default refraction coefficient for precise measurements. Furthermore, Equation (Equation 2) highlights that the vertical temperature gradient plays a dominant role in determining the refraction coefficient. As stated in Bomford [25], the intermediate atmosphere (heights of 20 m to 100 m above ground) maintains a relatively stable temperature gradient of about −0.01 K m^−1^, whereas the lower atmosphere experiences stronger variations. This stability in the intermediate layers is attributed to the limited influence of surface temperature on the atmospheric layers above [26].

Zhou et al. [27] suggest the use of reference points that experience similar atmospheric conditions as the measurement points and transfer of the atmospheric impact onto the additionally measured points. The work of Ehrhart and Lienhart [19] uses vertical angle observations towards a stable target to estimate the refraction coefficient for monitoring purposes. Additionally, Hennes [28], Eschelbach [29], and Reiterer [30] propose using turbulence measurements to estimate the refraction index gradient and angle of refraction for correcting vertical angle measurements, achieving a reduction in refraction effects by more than 75%. However, these methods, while useful for infrastructure monitoring, are not directly transferable to UAS tracking, where neither reference points nor stable targets can be placed in areas that experience similar influences as the UAS.

Based on these findings, it can be concluded that quantifying the effect of refraction on RTS measurements is challenging. The magnitude strongly depends on the vertical temperature gradient, which varies significantly over time. For airborne objects operating several hundred meters away from the RTS, the vertical temperature gradient along the line-of-sight may also vary due to differences in the underlying terrain. However, in the context of RTS-based trajectory measurements of a UAS, the UAS enables the acquisition of the vertical temperature gradient during the starting and landing phases and, consequently, the calculation of the refraction coefficient, according to Equation (Equation 2). Also, the majority of the measurement runs through the intermediate atmosphere, where the impact of refraction is expected to be smaller than in the lower atmosphere.

### 2.2. Systematic Deviations of 360° Prisms

The use of a 360° prism is essential to enable the RTS to track and measure a target regardless of its orientation. Several studies have explored the impact of systematic errors caused by Leica 360° prisms on the distance and angle measurement of the RTS [8,31,32,33]. These errors are caused by the smearing of the returning signals caused by the six individual facets and their transitions that make up the 360° prism. Systematic deviations of 360° prisms manufactured by other companies are investigated, for example, in Lackner and Lienhart [32]. They thoroughly investigate the cyclic errors associated with the use of 360° prisms, focusing on different models offered by Leica and Trimble. The study demonstrates the improvements achieved by the Leica GRZ-122 prism over the Leica GRZ-4 prism and further explores the cyclic effects in the smaller Leica GRZ-101 prism, which is of particular relevance for UAS tracking due to its significantly reduced weight and size. However, it is important to note that these studies were conducted for nearly horizontal sights with zenith angles close to 100 gon, leaving the effect of those errors for steep sights, e.g., at airborne targets, unanswered. The analysis of [32] reveals that the Leica GRZ-122 prism exhibits cyclic measurement variations with amplitudes of less than 2 mm in the horizontal component at a distance of 26 m, with the distance variation showing an amplitude of less than 1 mm. The vertical component exhibits minimal variation, below 1 mm. The smaller Leica GRZ-101 prism exhibits similar variations for the distance and horizontal components, but the vertical component experiences variations of approximately 2 mm. Since the cyclic nature of these effects is well understood, it is possible to correct for these variations if the orientation of the prism with respect to the RTS is known. As the measurements were carried out at 5 m and 26 m, it remains unclear how these effects change for increasing distances. Overall, they show that the influence of systematic deviations caused by the 360° prism on the RTS measurement is rather small and correctable. However, the change of the systematic deviations for steep angles and large distances has not been investigated yet. A thorough investigation of 360° prisms is necessary before using such prisms for trajectory measurements of a UAS. This investigation needs to include distances and vertical angles similar to those occurring when measuring the trajectory of a UAS. By correcting the systematic effects, the achievable accuracy of the RTS measurements can be increased, allowing for highly accurate measurement of targets using 360° prisms.

An alternative approach to using 360° prisms for platform tracking is proposed by [34]. In their work, an active reflector is developed. This system features a motorized platform that dynamically adjusts the prism’s orientation towards the total station. This active reflector requires real-time information about its position and orientation relative to the RTS. Using such a system, a more accurate circular round prism can be used instead of a 360° prism [35]. However, the active reflector introduces additional components, such as a motor and power source, which increase the weight of the reflector. In the context of UAS tracking, the added weight of the active reflector may pose practical challenges, along with the requirement for a data connection between the UAS and the RTS.

### 2.3. Kinematic Measurements of UAS with RTS

The RTS serves as the positioning component of the proposed decentralized multi-sensor system, dictating the achievable temporal and spatial resolution of the kinematic measurement task. In Stempfhuber and Sukale [8], several key challenges associated with kinematic object tracking using RTS are highlighted. These include (i) the inability to perform measurements in two faces of the RTS, a standard procedure for mitigating axis-related errors (e.g., collimation error) and enhancing the reliability of distance measurements. Another significant issue is (ii) the challenge of achieving precise prism alignment due to the prism’s continuous movement during tracking. Furthermore, (iii) the lack of accurate meteorological parameters impedes the effective correction of meteorological influences on measurements. Although (i) can be largely compensated for by a precise calibration of the RTS, (ii) relies on correction of the misalignment by signal processing techniques for automated reflector detection, such as ATR and ATRplus [21] or Autolock [22]. Meteorological parameters (iii) can be acquired by a sensor near the RTS, as in classical surveying, combined with a UAS-mounted sensor as described in Section 2.1.

The performance of modern RTS for measurements on moving targets has been extensively studied in Kälin et al. [6,36], Kerekes and Schwieger [37], Thalmann and Neuner [5], Stempfhuber and Sukale [8], Stempfhuber [33,38], Vogel et al. [39]. These articles examine various total station models and provide a basis for evaluating the applicability of using RTS to determine the trajectory of a UAS. Further research on the use of RTS to geo-reference kinematic multi-sensor systems is found in Vogel et al. [39], Brandstätter et al. [40], Hesse et al. [41], Liu and Noguchi [42], Vaidis et al. [43,44]. However, these studies primarily address terrestrial platforms with velocities smaller than 3 m s^−1^. The impact of insufficient time synchronization is less pronounced for such platforms. In addition, the movement characteristics and speeds are mostly different from those of the UAS.

Stempfhuber [33] systematically analyses the primary error sources in the tracking of kinematic objects, including the distance to the prism, the direction of movement relative to the total station, and the velocity of the prism. The particular impact of those factors on the measurement uncertainty depends largely on the RTS model used. Recent studies employ newer RTS models [5,8,37]. They show advances in RTS technology, as well as improved measurement frequencies and accuracies compared to older RTS models. However, the transferability of their results is limited, as their studies do not meet the velocities and ranges encountered in the context of UAS surveying.

Pan [45] focuses on the measurement of UAS trajectories using Leica TS60 RTS in conjunction with a network of six cameras to determine the UAS’s 6-DoF trajectory. Synchronization between instruments is achieved through an acoustic trigger signal for video timestamping. Furthermore, the internal clock and controller of the total station are synchronized using the correlation approach proposed by [46], which claims a maximum achievable synchronization error of less than 10 ms. Roberts and Boorer [47] use a Leica MS50 RTS for the trajectory measurement of a UAS. Synchronization with the control unit of a rotary motor is performed using a GeoCOM command to align the RTS clock with the controller clock. They report a synchronization quality of approximately 10 ms which results in a spatial uncertainty of approximately 2 cm, severely limiting the quality of the resulting trajectory. For their study, a modified 360° mini-prism is employed, featuring a downward-facing prism facet. This design enables measurements from directly below. In their work, no evaluation of the uncertainty induced by the customized prism is performed. Synchronization between the RTS and the camera mounted on the UAS is achieved in post-processing by cross-referencing the data. The accuracy of the UAS trajectory is assessed using ground control points, which reveal deviations of up to 3 dm. These deviations can be attributed to several factors, including the neglect of the lever arm, inadequate synchronization between the UAS camera and the RTS, potential inaccuracies introduced by the modified 360° mini-prism, and uncorrected time delays between the RTS subsystems.

Highlighting the relevance of UAS trajectories measured by RTS in different disciplines, the studies of Maxim et al. [48], Merkle and Reiterer [49], Paraforos et al. [50], Vougioukas et al. [51], and Vroegindeweij et al. [52] use RTS for UAS trajectory measurements. In Maxim et al. [48], UAS trajectories for architectural fabrication processes are measured using RTS. In [53], the accuracy of an RTS for kinematic observations is assessed using an industrial robotic arm. In their study, the robotic arm moves with a maximum speed of 1 m s^−1^, and the exhibited error of less than 1 cm is not further analyzed. Based on this study, Paraforos et al. [50] use an RTS to assess the accuracy of UAS-mounted imagery. In Vougioukas et al. [51] an RTS is used as a reference sensor to evaluate the localization of orchard workers relative to a vehicle using radio ranging, and Vroegindeweij et al. [52] use RTS to evaluate the localization of a robot in an aviary poultry house. Another notable contribution is given by Tombrink et al. [54] and Tombrink et al. [55], where a Leica TS60 is used to evaluate the accuracy of an inertial navigation solution on a moving multi-sensor system. This shows the importance of an extensive investigation of the uncertainty of RTS for UAS trajectory observations for a wide range of disciplines.

In Table 2 the most relevant contributions that investigate the achievable accuracies of UAS trajectory observations with RTS are compared. Publications that use the RTS as reference sensor to evaluate trajectories obtained by other technologies are not listed in this Table.

The direct measurement of UAS flight trajectories with RTS is addressed in Bláha et al. [56], and Pan [45], Roberts and Boorer [47], Maxim et al. [48], Kohoutek and Eisenbeiss [57]. However, the acquired trajectory is only critically discussed in Bláha et al. [56], Kohoutek and Eisenbeiss [57], and Roberts and Boorer [47], though none of these papers evaluate the accuracy of the acquired trajectory using adequate reference data. Adequate reference data should outperform the expected measurement uncertainty of the RTS and thus have a standard deviation of less than 1 cm for the 3D measurement. Additionally, these publications, except Pan [45], are more than nine years old. This means that the experiments were conducted with older RTS models, which are less capable than state-of-the-art models. Also, even though UAS are used, they are operated in a near-static mode to minimize any errors originating from the flight speed.

### 2.4. Current State of the RTS Measurement Process in the Context of UAS Trajectory Estimation

When looking at the existing literature that deals with RTS-based trajectory measurements of a UAS, the impact of refraction on the measurement, and the interaction of systematic prism deviations, we see the large relevance of these topics in the scientific community. However, up to now most publications focus on the measurement of terrestrial platforms [5,6,43]. Thus, for example, neither the effect of refraction for steep measurements of the trajectory of kinematic targets nor the change in systematic prism deviations of 360° prisms for steep measurements has been investigated. While some sources directly address the application of RTS for the trajectory estimation of a UAS, these sources are either outdated due to the rapidly advancing technology of RTS [47,56,57] or they do not investigate the achievable accuracy of the RTS measurements [48]. Other sources critically investigate the uncertainty budget of RTS for kinematic measurements [6,54] but restrict themselves to terrestrial close-range scenarios with rather slow-moving platforms. Consequently, there is a clear research gap that needs to be addressed to understand and model the process of RTS-based trajectory estimation of a UAS.

## 3. Time Synchronization of RTS

Ensuring proper time synchronization is critical when evaluating data from kinematic multi-sensor systems. This includes time synchronization not only between segments, such as the ground and kinematic segments, but also within each sensor system. For example, an RTS, which incorporates more than 17 individual sensors [58], exhibits the character of a multi-sensor system in itself. Since the investigations conducted by Stempfhuber [33], it has been well-established that one of the most critical time synchronization challenges in a total station arises between the EDM and the angle measurement sensors.

### 3.1. Temporal Calibration of RTS

Based on these findings, Thalmann and Neuner [5] distinguish two types of synchronization parameters. The intrinsic parameter refers to the delay within the RTS itself, specifically between the EDM unit and the angle measurement sensors. In contrast, the extrinsic parameter pertains to delays between the measurements acquired by the RTS and their reception by a controlling unit, where the data are timestamped in a reference time system. This parameter also includes the interface latency. Both synchronization parameters must be addressed for the acquisition of accurate data about kinematic platforms using RTS. According to Thalmann and Neuner [5], intrinsic timing delays of several milliseconds have been observed for Leica TS16 total stations. For platforms moving at speeds around 10 m s^−1^, such delays can lead to positional errors of several centimetres, depending on the direction of movement. In contrast, extrinsic synchronization involves the incorrect association of a timestamp with a position. In a multi-sensor system, this can result in misaligned UAS measurements and significant errors in the geo-referenced data. Thalmann and Neuner [5] identify extrinsic latency parameters in the range of 70 ms, which, even for slow-moving platforms (e.g., 1 m s^−1^), introduce errors of approximately 7 cm. To ensure accurate absolute kinematic measurements, these delays must be corrected. Significant improvements in intrinsic synchronization have been reported for the Leica MS60 model, which incorporates ATRplus technology and a more advanced electronic distance measurement unit [7,59]. Evaluations by Stempfhuber and Sukale [8] and Grimm and Hornung [7] confirm that the intrinsic latency parameter of the MS60 model is reduced to levels below the instrument’s measurement accuracy. However, these tests were performed at velocities under 3 m s^−1^. Since the impact of inadequate intrinsic synchronization is directly proportional to the lateral velocity of the observed object, higher velocities may still produce noticeable errors. Furthermore, uncorrected extrinsic latency continues to be a major source of error in geo-referencing multi-sensor systems. In their research, Thalmann and Neuner [5] introduce an approach for the temporal synchronization of RTS, addressing both intrinsic and extrinsic time delays. This method utilizes the circular horizontal motion of an industrial robot arm as a highly accurate reference system. The approach builds on previous findings of Depenthal [60], who explored the feasibility of using controlled horizontal movements as a reference to estimate time delays within RTS. Furthermore, Thalmann and Neuner [5,61] propose a method for synchronizing the measurement data of modern RTS with an external controller. This external controller is synchronized to a reference time frame using the network time protocol (NTP) over wired and wireless connections. A similar approach is implemented by Kerekes and Schwieger [37] for the synchronization of multiple RTS. In a related development, Kälin et al. [6] utilize the precise time protocol (PTP) to synchronize a controller for RTS measurements to a reference time frame. PTP provides a more precise synchronization alternative, particularly advantageous in applications requiring sub-millisecond accuracy, further enhancing temporal alignment within decentralized systems.

The temporal calibration procedure of Thalmann and Neuner [5] consists of three main steps:Controller Synchronization: Estimation of the temporal offset $dt^{c}$ and the frequency error $f^{c}$ between the external controller and a time reference.Sensor Synchronization: Estimation of the temporal offset $dt^{s}$ and frequency error $f^{s}$ between the time-referenced controller and the sensor board of the RTS, which combines the measurement data of the individual submodules.Temporal Calibration: The final step focuses on estimating the intrinsic $\delta t^{a}$ and extrinsic $\delta t^{d}$ latency (which includes the interface latency $\delta t^{i}$) of the RTS, using the reference sensor (robotic arm).

These three synchronization steps are visually represented in Figure 2.

The work of Thalmann and Neuner [5] also thoroughly evaluates the synchronization quality achieved through the three steps of the temporal calibration routine by estimating the remaining synchronization uncertainties. The study concludes that the synchronization of two RTS can be achieved with a remaining uncertainty of approximately 0.2 ms. Kerekes and Schwieger [37] presents a similar value of 0.3 ms for the synchronization quality of a network of multiple RTS. The residual uncertainty found by Thalmann and Neuner [5] is attributed to three primary sources:The effect of controller synchronization using NTP contributes about 40 μs to the overall uncertainty.The uncertainty associated with sensor board synchronization is approximately 70 μs.The uncertainty of extrinsic latency, denoted as $\delta t^{d}$, adds around 80 μs to the total uncertainty.

It is important to note that these values were obtained under controlled laboratory conditions, which may not fully reflect the complexities of real-world environments, such as fluctuating temperatures.

In conjunction with the temporal synchronization approach, Thalmann and Neuner [5] introduce a model for computing 2D Cartesian coordinates in kinematic scenarios (Equation (Equation 3)). This model incorporates the intrinsic time delay, $\delta t^{a}$, and the extrinsic time delay, $\delta t^{d}$, of the RTS. In addition, it accounts for the distance rate (*v*) of the measured distance (*D*) and the angular rate (ω) of the observed horizontal angle *R*. By integrating these factors, the model accurately represents the displacement vector induced by the intrinsic and extrinsic time delays of the RTS. Figure 3 illustrates this displacement vector arising from the intrinsic and extrinsic time delays of the RTS.

With Equation (Equation 3), it is possible to calculate the coordinates *x* and *y* for a time $t_{j}$ using the parameters estimated in the temporal calibration as well as the measured distance *D* at time $t_{d}$ and the measured horizontal angle *R* at time $t_{a}$. The static coordinates of the RTS are denoted as $x_{0}$ and $y_{0}$.(3)$$\begin{array}{cc} x(t_{j}) & =x_{0}+\left[D\left(t_{d}\right)+v_{d}\left(t_{d}\right)\;\delta t^{d}\right]\cos\left[R\left(t_{a}\right)+\omega \;\cdot \;t_{a}\left(\delta t^{d}+\delta t^{a}\right)\right]\\ y(t_{j}) & =y_{0}+\left[D\left(t_{d}\right)+v_{d}\left(t_{d}\right)\;\delta t^{d}\right]\sin\left[R\left(t_{a}\right)+\omega \;\cdot \;t_{a}\left(\delta t^{d}+\delta t^{a}\right)\right] \end{array}$$

The presented method provides a promising approach to mitigate the effects of time delays exhibited by RTS during kinematic surveys. However, the given model, as well as the calibration procedure, are restricted to the two-dimensional space. The intrinsic delay of the vertical angle remains uncorrected, introducing uncertainties, particularly for the measurement of airborne multi-sensor systems.

Research on RTS synchronization was also conducted by Gojcic et al. [62], focusing on the synchronization of multiple RTS and the impact of temperature fluctuations on the internal time frame of the instruments. Their approach involves using a connected GNSS receiver to evaluate the internal time accuracy of a total station at varying temperatures. Laboratory experiments were performed for temperature calibration, modeling the time drift based on time offset measurements with a temperature interval of 10 K across a range of 0 °C to 50 °C. From these measurements, a cubic polynomial calibration function is derived to correct the internal clock. The results show that applying this temperature drift calibration function reduces the internal time error of the investigated Leica MS60 RTS relative to the GNSS time to 12 ms after 12 h. In contrast, without calibration, the error is significantly higher, being 2.5 s. For the synchronization of multiple total stations, Gojcic et al. [62] employs a cross-correlation technique using measurement data from a commonly measured movement, such as an up-and-down prism trajectory. By aligning the minimum and maximum points in angle measurements, the time delays are estimated. The accuracy of this estimation is limited by the RTS sampling frequency of approximately 20 Hz. The study concludes that the calibration routine achieves a synchronization error of less than 50 ms between two total stations, even eight hours after the calibration procedure. However, changes from the estimated constant time drift are observed, reaching around 4 ppm across the total station’s temperature range. Such drifts can significantly affect the measurement of kinematic targets and must be corrected for highly accurate applications. It is important to note that the temperature calibration performed in the study is insufficiently resolved for the entire temperature operating range of the total station. The authors suggest using smaller temperature steps within the calibration routine to improve the accuracy of temperature-dependent time drift corrections. This calibration routine is particularly valuable for scenarios in which continuous synchronization to a reference time frame, as described in Thalmann and Neuner [5], is infeasible. However, when a connection to a reliable time reference is available, continuous synchronization should be prioritized over this correlation and temperature-based calibration, as it offers superior accuracy.

### 3.2. Recent Work on Time Synchronization of RTS in the Context of Kinematic Measurements

In this subsection, we review recent research on using RTS for kinematic measurements. In Vogel et al. [39], a Leica TS60 RTS measures a reflector moving at 0.07 m s^−1^. The control of the RTS is based on an application provided by the manufacturer, namely Leica Captivate TPS Survey Streaming, optimized for measuring moving objects. They compare the RTS measurements with timestamped laser tracker data and estimate the temporal offset of the RTS. The low velocity and measurement distance make it difficult to transfer the results to UAS measurements. However, they come to a similar conclusion as Thalmann and Neuner [5], that an extrinsic latency of about 55 ms needs to be considered to time reference the measurements of an RTS. Kälin et al. [6] use a decentralized multi-sensor system comprising an RTS and motion capture cameras to determine the 6-DoF pose of a car. The system uses the Precision Time Protocol (PTP) to achieve a three-sigma synchronization accuracy of 1 ms. The standard deviation of the RTS measurement, which has a sampling frequency of 20 Hz, is stated with approximately 4 mm for speeds between 1 and 2 m s^−1^, but increases up to 9 mm for speeds of 4 m s^−1^ to 5 m s^−1^. In Kälin et al. [36], the performance of a Leica MS60 RTS is assessed using a high-resolution motion capture system. Both systems measure a toy train that moves at about 0.6 m s^−1^, and the authors report a successful integration of RTS and motion capture data. The time synchronization between both systems is realized by using cross-correlation of the measurements in post-processing. In conclusion, they report a verification setup for various sensor investigations. However, the motion capture system restricts the test range to the laboratory. Another contribution is the study by Vaidis et al. [43], in which three RTS are used to measure three prisms mounted on a vehicle to determine its 6-DoF pose. This approach, aimed at replacing the GNSS-RTK positioning in areas with poor GNSS reception, claims a precision of up to 10 mm in position and 0.6° in orientation. However, with a measurement rate of 2.5 Hz, the system requires low vehicle velocities to achieve a dense trajectory. Due to the low velocity, the effect of synchronization errors is also reduced. The performed evaluation is based on GNSS-RTK measurements, whose uncertainties overlap with the reported precision. This raises questions about the reliability of the stated results. In their study, synchronization and data processing are managed via radio communication with Raspberry Pi 4 computers attached to the instruments.

Kerekes and Schwieger [37] evaluate the synchronization and measurement uncertainties of four total stations triggered by a single controller against the measurements of a laser tracker. They report RMS values for individual RTS measurements less than 5 mm before and 2 mm after outlier elimination. In addition to the detailed investigation of the synchronization of RTS, Thalmann and Neuner [5] also investigated the accuracy achieved by the calibrated Leica TS16. For circular motion at 25 m distance with a velocity of about 3 m s^−1^, they conclude a standard deviation of about 2 mm for the distance measurement and for the angle measurement. These values come close to the values given in the manufacturer’s data sheet for a static measurement. Further investigations of the application of multiple RTS for the measurement of kinematic platforms are presented in Kerekes and Schwieger [63,64]. In Kerekes and Schwieger [63], simultaneous observation of a moving platform using multiple RTS is explored, motivated by the limitations of single-RTS tracking when line-of-sight obstructions occur. The use of two RTS significantly enhances the robustness against obstructions and improves the accuracy of the measured position. The improvement in accuracy is attributed to the high accuracy of angular measurements at short distances, which minimizes errors when using well-conditioned intersection angles. However, this accuracy advantage diminishes at larger distances. The study reports a reduction of the Helmertian error from nearly 5 mm (single RTS) to less than 2 mm (dual RTS). Reference [64] focuses on the angle measurement capabilities of RTS, noting that the angle measurement rate exceeds the combined distance and angle measurement rate and delivers superior accuracy at short distances. In this study, two RTS perform a spatial forward intersection on a moving reflector, relying on the angular measurement for position determination. Similar to Kerekes and Schwieger [63], synchronization of the two RTS is not explicitly addressed. The setup for this investigation is illustrated in Figure 4.

The evaluation of the measurements focuses on the lateral distance from a straight line along which the prism moves. The experiment reveals average lateral deviations of 2–3 mm, with maximum deviations of up to 7 mm. Although simulations predicted an accuracy below 1 mm, observed deviations exceed these estimates. However, their studies emphasize the potential improvement of position accuracy by using the angle measurements of multiple RTS. Their main motivation is the increased measurement frequency of angle measurements, which is only valid for specific RTS models. For example, the Leica MS60 RTS model achieves similar measurement frequencies for angle and combined distance and angle measurements.

### 3.3. Current State of Time Synchronization for RTS

In the context of time synchronization, there has been a vast increase in publications that utilize RTS for time-synchronized measurements in recent years. The availability of the PPS signal of GNSS systems makes synchronization of distributed sensors possible, and data from different sensors can be merged. There are also very thorough analyses on the achievable accuracies when using time-synchronized RTS measurements of moving targets. Those studies agree that, in general, the RTS can be synchronized to a time reference with an uncertainty of less than 1 ms, and the geometric measurement performance is on the very low millimeter level. However, these investigations are almost exclusively performed under laboratory conditions [5,37,39] and are thus limited in terms of range, impact of atmospheric refraction, and platform velocity. Therefore, the transferability of the obtained insights to outdoor environments is limited, and the extension of those studies into a realistic outdoor environment is necessary and of great value for a variety of fields in addition to geodesy, e.g., precision farming or robotics.

## 4. Image-Based Total Station Measurements

IATS extend a classic RTS with imaging capabilities. Some RTS have imaging capabilities installed by the manufacturer (e.g., Leica MS60). However, a custom camera mount on the RTS, e.g., on the ocular, turns every RTS into an IATS. This approach is not unusual and is also used in Wagner et al. [14], Bürki et al. [65], Guillaume et al. [66], and Hauth et al. [12]. A review on image-assisted total stations in regard to structural health monitoring is conducted in Zschiesche [67]. Custom-built IATS offer the advantage that a suitable camera can be selected for a specific application. In addition, cameras installed in commercially available IATS are easily outperformed in terms of resolution and image frequency by standard cameras, e.g., those manufactured by GoPro. For example, Svanström et al. [68] use a thermal camera in combination with an RGB camera in an IATS to detect a UAS. Provided that the camera itself and its geometric relation to the total station are precisely calibrated, the IATS can leverage the images for photogrammetric measurements. Research in this field demonstrates promising results for accurate coordinate determination based on image data. Another contribution is the simulation study by Niemeyer et al. [15] that assesses the estimation of the pose of a UAS using IATS. In their work, the RTS part of the IATS measures the position of a prism mounted on a UAS. In addition, the UAS has photogrammetric markers installed that are measured in the image taken by the IATS. These markers enable the derivation of the orientation of the UAS, complementing the prism’s position data. Consequently, the IATS measures the 6-DoF pose of the UAS without additional sensors. Their study explores two methods: single-image photogrammetry and stereo-photogrammetry using additional cameras. In single-image photogrammetry, the prism’s position and marker measurements are combined to estimate the orientation of the UAS. The authors report a total point error below 5 mm at distances up to 100 m, strongly depending on the characteristics of the IATS (measurement accuracy, calibration accuracy, etc.). The stereo-photogrammetric approach uses additional cameras and achieves higher orientation precision. A Monte Carlo simulation reveals this method’s superiority over the single-image photogrammetry, as the resulting measurement accuracy depends less on the position of the UAS. However, for most constellations, the accuracy of the marker position is identical for both approaches. With 10 markers on the UAS, the rotational parameters can be determined with a standard deviation of 0.15° to 0.30° and the marker positions with a less than 1 cm standard deviation. This is valid for most investigated conditions using either of the two approaches. However, the study does not consider factors such as atmospheric refraction or deviations of the IATS axis, incorrectly favoring steep sightings for maximum accuracy. In addition, mounting 10 or more markers on a UAS seems ambitious for a real setup, and the results might be significantly worse for a smaller number of markers. In conclusion, the simulation study of Niemeyer et al. [15] identifies key accuracy factors such as the position of the UAS, angle measurements, image coordinates, and camera constants for the IATS-based orientation estimation but is limited to a theoretical approach.

In Ehrhart [13], Ehrhart and Lienhart [19,69,70], Lienhart et al. [71] the on-axis telescope camera of a modern IATS is analyzed and used for various measurement tasks. A motivation for their research is the fact that the image measurements enable the measurement of natural targets without the need for retro-reflective prisms. In Ehrhart [13], the performance of the on-axis telescope camera of a modern IATS is analyzed, with a detailed discussion of the error sources and the relationship between the image coordinates and angle measurements. The study concludes that, under laboratory conditions, an accuracy of approximately 0.1 mgon can be achieved for image-based measurements. To achieve this, the authors performed a calibration of the telescope camera over all possible focus settings of the IATS. In addition, they emphasize the need for a 3 h warm-up period for the IATS before measurements are made, as the warm-up effect can introduce an error of about 1 mgon. Their research also emphasizes that, in order to minimize systematic errors (e.g., radial distortion), the target should be positioned near the principal point of the image sensor. In addition, they state that performing two-face measurements can significantly enhance accuracy and mitigate warm-up effects.

## 5. Integrated Trajectory Estimation

A multi-sensor system incorporates additional sensors to the RTS, typically an IMU to measure the orientation of the system and to improve the trajectory and a mapping sensor such as a UAV-borne laser scanner (ULS) or camera. The conventional workflow of multi-sensor systems uses GNSS and IMU data to estimate a 6-DoF trajectory [72]. This trajectory is then used to perform direct geo-referencing of the data acquired by the mapping sensor [73].

The trajectory estimation typically employs Kalman filtering [72] to fuse the GNSS and IMU data in order to produce a 6-DoF trajectory. While for a loosely coupled Kalman filter, the position data of the GNSS can be replaced with position data from the RTS, Thalmann and Neuner [16] present a tightly coupled Kalman filter designed for the fusion of RTS and IMU data. After direct geo-referencing, the overlap of the mapping data is used to minimize remaining discrepancies, estimate lever arms and boresight angles, and improve the trajectory [18,74,75]. For ULS, this process is referred to as strip adjustment, and for camera data it is referred to as bundle block adjustment. Instead of separating the steps of trajectory generation and leveraging the mapping data, current research investigates direct use of the mapping data in trajectory estimation [2,76,77,78,79]. For example, Cucci et al. [80] employ dynamic networks, as introduced by Colomina and Blázquez [81], to integrate GNSS, IMU, and image observations in a tightly coupled trajectory estimation. A key advantage of this method is the straightforward implementation of tie points using static spatial constraints. Moreover, this approach consolidates the evaluation of GNSS, IMU, and image observations into a single processing step, simplifying the processing pipeline for such sensor systems. However, this unified approach increases computational demands due to the added complexity of simultaneously handling a larger number of parameters. Importantly, this method not only improves the overall 6-DoF trajectory estimation but, in particular, enhances the accuracy of the attitude estimation. The approach can be transferred to the use of ULS point-to-point correspondences in trajectory estimation. Brun et al. [17] describe a method that incorporates 3D correspondences as static constraints within the dynamic network, similar to the use of image observations. The concept of 3D correspondence-based trajectory improvement, as presented in Brun et al. [17], is illustrated in Figure 5. Their findings demonstrate that even a small subset of available 3D correspondences, such as 1% or even 0.1%, can significantly enhance the estimation of the trajectory. Furthermore, using approximately 5% of the 3D correspondences yields improvements comparable to those achieved with the full dataset of correspondences. Their research uses low-cost MEMS inertial sensors, leaving a larger potential for improvement of the trajectory than, e.g., the use of a high-accuracy IMU. This shows the practical relevance of the presented methodology, as low-cost MEMS inertial sensors are commonly used in UAS surveying.

Pöppl et al. [18] presents an alternative method for the estimation of trajectory with airborne laser scanning (ALS). The approach integrates position data of GNSS, IMU data, and mapping data in a holistic least-squares adjustment. This integrated trajectory estimation incorporates redundant observations, specifically plane observations, derived from mapping sensors in object space alongside on-board IMU and GNSS data. Using all available information in a single least-squares adjustment, this integrated trajectory estimation improves accuracy and robustness. Compared to the traditional direct geo-referencing workflow with subsequent trajectory correction using mapping sensor data [73,82], the integrated trajectory estimation shows significant improvements. Pöppl et al. [18] report approximately a 50% improvement in GNSS, IMU, and ALS adjusted data sets compared to conventional Kalman filtering methods. Their method loosely couples GNSS and IMU data, which enables replacing the position data of GNSS with RTS observations, making their method directly applicable for RTS-based multi-sensor systems. However, while Kalman filtering supports real-time applications, integrated trajectory adjustment is a post-processing solution that aims at maximizing accuracy. In [20], the challenge of integrating data from both airborne LiDAR and photogrammetry in trajectory estimation is addressed. Rather than solving LiDAR strip adjustment and aerial triangulation independently, which can result in discrepancies of several decimeters between both data sets, a rigorous integration of both mapping data streams is proposed, termed hybrid orientation of LiDAR point clouds and aerial images. The authors conduct an experiment to evaluate the performance of this hybrid method. One notable result is the significant improvement in the roll angle correction function. By incorporating image data, the LiDAR-based roll angle correction is enhanced, leading to increased block stability and reduced deformations in the resulting model. Importantly, rigorous hybrid adjustment minimizes discrepancies between the image-based and LiDAR-based data, greatly facilitating the joint analysis of these datasets. A comprehensive review of trajectory estimation methods for kinematic platforms is presented in Pöppl et al. [3], with a particular focus on kinematic mapping. The review emphasizes the integration of GNSS, INS, and mapping sensors, such as laser scanners and cameras, in the estimation of the 6-DoF trajectory. The study highlights the multidisciplinary nature of three-dimensional trajectory estimation, where advances in fields such as robotics, geodesy, and computer vision have contributed significantly. For instance, continuous trajectory representation methods, developed in robotics, offer advantages over discrete-state approaches when integrating multiple sensors with different sampling frequencies and asynchronous triggers. The review article also underscores the transformative role of increased computing power in expanding trajectory estimation methodologies. This advancement supports a shift from trajectory-level error modeling, which can introduce deformations of the trajectory and consequently the 3D model, to sensor-level error modeling. The latter adopts a holistic multi-sensor estimation approach, where errors are modeled directly at the sensor level. A prominent example of this paradigm is the tightly coupled evaluation of multi-sensor systems. Figure 6 and Figure 7 illustrate the differences between trajectory-level and sensor-level error modeling, respectively. The authors claim that tightly coupled estimation offers theoretical advantages over loosely coupled approaches. However, its primary drawback lies in the increased complexity of the model. This complexity requires a detailed knowledge of sensor-specific error characteristics, access to raw data, and higher computational resources [83]. The authors also note that many approaches are experimentally evaluated with specific sensors and applications. This limits the generalizability of the algorithms and results, as differing sensor configurations and application scenarios often hinder the applicability to certain use cases. Furthermore, the absence of high-accuracy reference trajectories makes comprehensive comparisons between different methods challenging.

## 6. Conclusions and Outlook

Airborne, fast-moving UAS introduce additional challenges for RTS tracking that are not present for terrestrial platforms. In this literature review, we show that RTS are already used for time-synchronized measurements to kinematic platforms and can achieve mm-accuracy for velocities below 3 m s^−1^ and ranges below 50 m. While RTS are also already used for measurements of UAS, a thorough accuracy investigation does not exist. In addition, several influence factors, e.g., atmospheric refraction and systematic deviations of 360° prisms, are well researched for terrestrial applications; here, the adaptation and extension to UAS measurements is missing. Also, the estimation of UAS orientation from IATS images has been researched but not practically implemented and validated. Furthermore, state-of-the-art integrated trajectory estimation algorithms focus on GNSS/IMU integration but do not consider RTS as a positioning sensor.

Thus, the current state-of-the-art provides starting points for methodologies to calibrate and time-reference RTS measurements and enable sensor fusion with IMU data to allow for accurate trajectory estimation. Modern IATS even allow for the extension of the 3D position measurement with orientation information of the UAS. Recent advances in integrated trajectory estimation also make use of mapping sensor observations in the trajectory estimation process, leading to higher accuracies than the conventional integration of position observations and inertial measurement data.

In general, we show that many theoretical foundations for the described decentralized multi-sensor system (Section 1) are laid, but they all require further development and adaptation to achieve our desired accuracy of sub-cm and sub-0.01 gon for the RTS-based trajectory estimation of a UAS. In addition, we show the large demand for accurate reference trajectories, which RTS or IATS can offer, for the development of sensors and methods for trajectory estimation.

Based on Figure 1 and Table 1, we can identify the most important research steps to realize a decentralized multi-sensor system based on IATS to estimate UAS trajectories. It is necessary to (i) extend the time synchronization and temporal calibration approach of Thalmann and Neuner [5] as well as the uncertainty investigations of Group A to larger velocities, up to 10 m s^−1^, longer ranges, up to 500 m, and more recent RTS models. The impact of atmospheric refraction (ii) needs to be accurately estimated for kinematic measurements of UAS trajectories, and a feasible strategy to minimize this impact needs to be created (Table 1, Group B). Moving on to image-based aspects, (iii) the evaluation of image-based measurement with IATS needs to be extended through the aspect of time synchronization for successful integration in a decentralized multi-sensor system (Table 1, Group C), and (iv) the orientation estimation of a UAS, proposed in Niemeyer et al. [15] (Table 1, Group D), must be practically realized and proven in addition to having its uncertainty estimated. Finally, (v) the trajectory estimation strategies as presented in (Table 1, Groups E and F) need to be adapted for their application with a UAS and RTS. For all these steps, the required research must address both the functional and stochastic relations. Ultimately, we will obtain a complete framework for the trajectory estimation of decentralized multi-sensor systems based on RTS with the corresponding uncertainty assessment.

The aim in our future work is to address exactly these research gaps and leverage the large potential that lies in RTS-based multi-sensor systems. We want to show the capabilities of modern RTS and IATS for trajectory measurements, using the example of a UAS, first based on a comprehensive simulation framework and then on field measurements. For this, we need to refine and extend existing approaches, e.g., temporal calibration routines, IATS image measurements, and integrated trajectory estimation for our RTS and UAS use case. Finally, we will obtain a precise stochastic model of IATS-based trajectory estimation and integrate these observations with the onboard sensors of a UAS using an integrated trajectory estimation framework to acquire cm-accurate point clouds based on a decentralized, RTS-based, multi-sensor system.

## Figures and Tables

**Figure 1 sensors-25-03838-f001:**
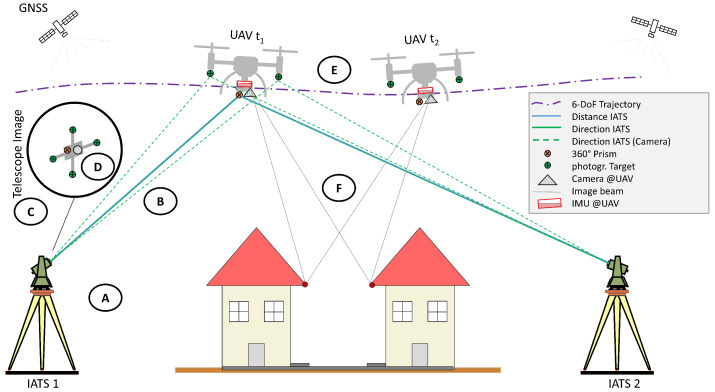
Decentralized multi-sensor system: Ground segment consisting of multiple IATS and kinematic segment of UAS with IMU, GNSS antenna and receiver, and mapping sensor, e.g., a camera. The letters refer to Table 1, where A relates to the IATS synchronization process and data recording, B to the interaction of the measurement with the atmosphere, C to the camera image acquisition and D to the processing of the acquired image for the orientation estimation. Letter E relates to the trajectory estimation using IMU data in addition to the IATS measurements and F relates to the integrated trajectory estimation using IATS, IMU and point cloud information in the form of correspondences.

**Figure 2 sensors-25-03838-f002:**
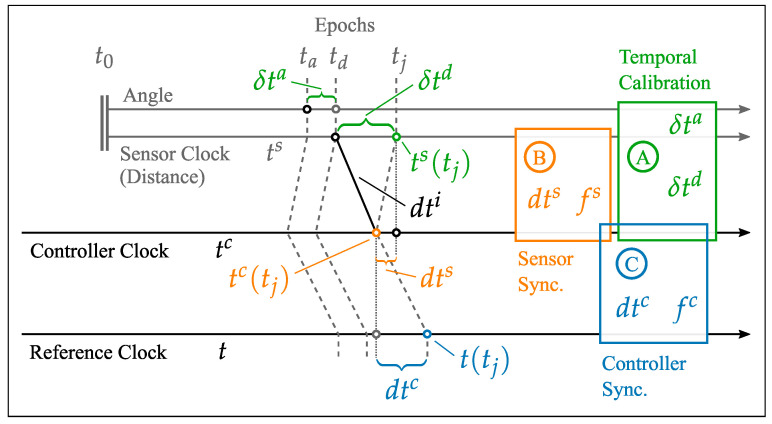
Steps of temporal calibration routine [5].

**Figure 3 sensors-25-03838-f003:**
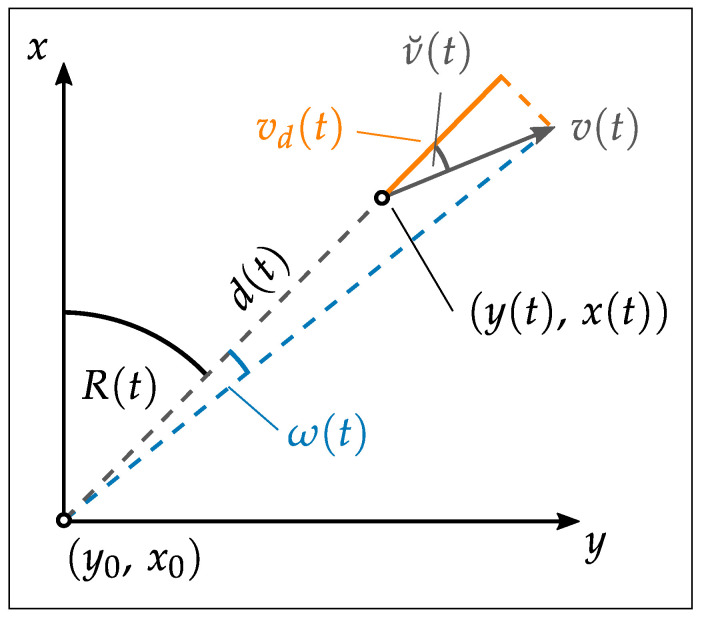
Kinematic formulation of polar measurements using time-dependent azimuth R(t), distance D(t), and their derivatives $\omega \left(t\right)=\frac{dR}{dt}$ and $v_{d}\left(t\right)=\frac{dD}{dt}$. The angle between the line of sight and the moving direction is denoted by $\overset{˘}{v}\left(t\right)$ [5].

**Figure 4 sensors-25-03838-f004:**
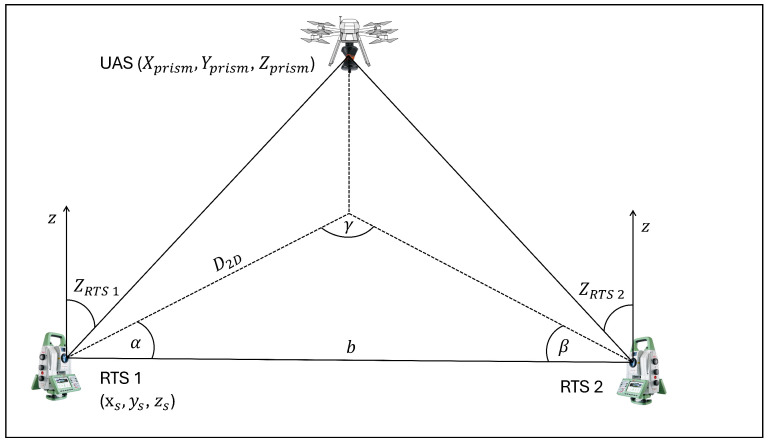
Forward intersection with two RTS, adapted from [64].

**Figure 5 sensors-25-03838-f005:**
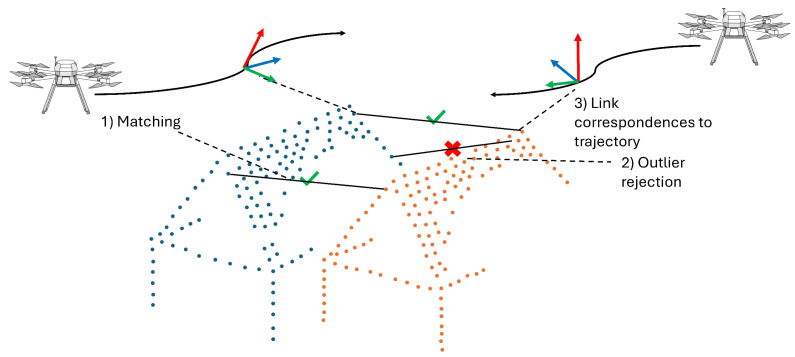
Principle of trajectory connection based on 3D correspondence detection. Two misaligned LiDAR point clouds (blue and orange) are related to two overlapping flight lines, adapted from [17].

**Figure 6 sensors-25-03838-f006:**
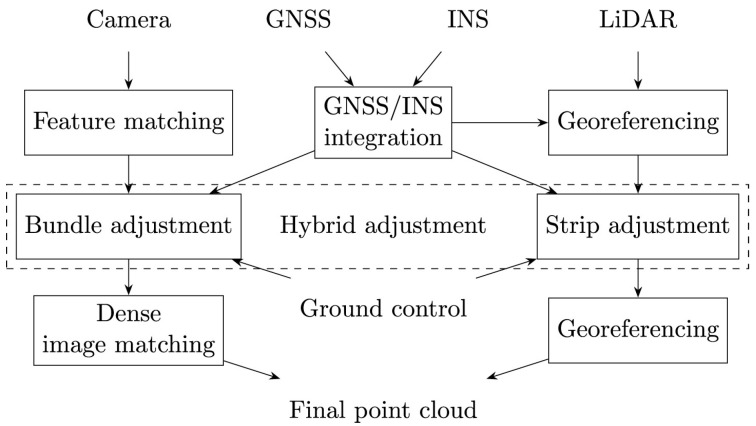
Standard kinematic mapping processing pipeline with trajectory-level error modeling [3].

**Figure 7 sensors-25-03838-f007:**
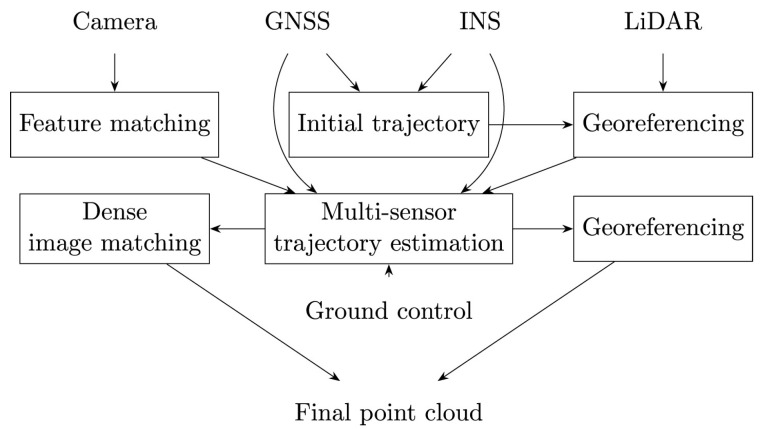
Holistic kinematic mapping processing pipeline with sensor-level error modeling [3].

**Table 2 sensors-25-03838-t002:** Contributions that thoroughly investigate the achievable accuracies of UAS trajectory observations with RTS.

Publication	Adequate Reference ($\sigma_{3D}<1$ cm)	Reference	Speed	Year
Bláha et al. [56]	No	DGNSS	Not stated	2012
Kohoutek and Eisenbeiss [57]	No	DGNSS	Hovering	2012
Roberts and Boorer [47]	No	Photogrammetry	Hovering	2016

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
