# Peer review of "A Review on UAS Trajectory Estimation Using Decentralized Multi-Sensor Systems Based on Robotic Total Stations"

_sensors, 2025, doi:10.3390/s25133838_

Round 1
Reviewer 1 Report
Comments and Suggestions for Authors
This article is a review on decentralized multi-sensor systems based on robotic total stations. Then the following problems exist:
In the abstract section of a review paper, it is necessary to briefly describe how the review is conducted, what methods are used, how the total station problem of a distributed multi-sensor system is reflected, what problems are found after the review, what are the advantages and disadvantages, and provide a brief explanation and summary. This is a review. It is suggested that the author make some modifications
2. The article first divides the review of the core issue into two key points: a review and analysis of the existing research related to trajectory estimation of moving platforms in RTS and IATS. In the INTRODUCTION section, the key literature read and the literature analysis should be presented and explained in the form of a table, followed by the innovative points.
3. The explanation of the innovation points should be presented in a separate paragraph instead of being directly followed by the main text.
4. The second part, "Measurement process of RTS", is all a literature review and there is no analysis of the author's reading of these literatures.
5. The third part also lacks analysis
6. There are issues with the layout of the entire thesis. The paragraphs are not divided correctly and there are many format errors.
7. The conclusion section states that the relevant theories based on the estimation of autonomous flight trajectories of unmanned aerial vehicles need further development and adjustment. The conclusion is too general and could have been reached without this review, failing to reflect the significance of the author's discussion.
8. The entire text is mainly argumentative rather than a review. A review requires analysis; a better review would raise questions and provide arguments. This article neither addresses any points nor conducts any confirmatory analysis, thus having a relatively low level of support.
The conclusion proposes that the future goal is to address the research gap. Does the gap refer to the reliability of the uncertainty budget for motion observation in RTS? There is considerable room for improvement. It is suggested that the author make thorough revisions.
Author Response
Comment 1: This article is a review on decentralized multi-sensor systems based on robotic total stations. Then the following problems exist:
Answer 1: Dear Reviewer,
thank you for the very detailed and constructive feedback. We sincerely enjoyed your feedback in order to significantly improve our work.
Comment 2: In the abstract section of a review paper, it is necessary to briefly describe how the review is conducted, what methods are used, how the total station problem of a distributed multi-sensor system is reflected, what problems are found after the review, what are the advantages and disadvantages, and provide a brief explanation and summary. This is a review. It is suggested that the author make some modifications
Answer 2: We admit that the abstract is very compact and an extension of the abstract improves the paper, as the abstract is a very important part of the paper. Thus, we extended the abstract significantly.
Comment 3: The article first divides the review of the core issue into two key points: a review and analysis of the existing research related to trajectory estimation of moving platforms in RTS and IATS. In the INTRODUCTION section, the key literature read and the literature analysis should be presented and explained in the form of a table, followed by the innovative points.
Answer 3: Thank you for pointing out the weak analysis part of our work. We tried to add those substantial parts of a review to our work. As part of that, we followed your suggestion and added the six key publication groups, which we identified during our review, to the introduction. They are now summarized in a Table, together with their main contributions from the perspective of UAS trajectory estimation in decentralized multi-sensor systems based on robotic total stations. In the Conclusion part we go back to this Table and derive very detailed research gaps triggered by these key publications.
Comment 4: The explanation of the innovation points should be presented in a separate paragraph instead of being directly followed by the main text.
Answer 4: We elaborate on the Table and thus the contributions / innovation points of the key publications in a paragraph following the Table. We hope this sufficiently addresses this point.
Comment 5: The second part, "Measurement process of RTS", is all a literature review and there is no analysis of the author's reading of these literatures.
Answer 5: For the second part we added a whole subsection analysing the previously presented work and putting it into the correct context.
Comment 6: The third part also lacks analysis
Answer 6: Similar to the second part, we added a whole subsection to the third part where we analyse the previously presented work and put it into the correct context.
Comment 7: There are issues with the layout of the entire thesis. The paragraphs are not divided correctly and there are many format errors.
Answer 7: We discussed this issue within the author group and carefully revised the document to fit with the MPDI Style guidelines as well as proper formatting. As we use the MDPI Sensors LaTeX style sheet we do not have many options to influence the formatting. However, without more information or examples we do not understand what is meant by “paragraphs are not divided correctly” or “many format errors”. Please elaborate on this, and we are happy to revise the document accordingly.
Comment 8: The conclusion section states that the relevant theories based on the estimation of autonomous flight trajectories of unmanned aerial vehicles need further development and adjustment. The conclusion is too general and could have been reached without this review, failing to reflect the significance of the author's discussion.
Answer 8: We revised the entire conclusion, leading to a more detailed conclusion. It now reflects most insights gained during our literature review and.
Comment 9: The entire text is mainly argumentative rather than a review. A review requires analysis; a better review would raise questions and provide arguments. This article neither addresses any points nor conducts any confirmatory analysis, thus having a relatively low level of support.
The conclusion proposes that the future goal is to address the research gap. Does the gap refer to the reliability of the uncertainty budget for motion observation in RTS? There is considerable room for improvement. It is suggested that the author make thorough revisions.
Answer 9: This is a general remark on our work, which summarizes the previously mentioned points. We are confident to have improved our work based on the provided comments.
Thank you,
The Authors
Reviewer 2 Report
Comments and Suggestions for Authors
The suggestions for modification are as follows:
1.It is recommended to include the current domestic and international research status of distributed multi-sensor systems for robotic total stations and to conduct comparative research analysis.
2.It is recommended to make certain adjustments to the structure of the article. Currently, a large portion of the article is focused on describing RTS technology. Adding comparative analyses of related technologies could highlight the advantages of this technology.
- It is recommended to add some of the latest references in this field, which can be used for comparison with RTS technology.
Author Response
Dear Reviewer,
thank you for the feedback. It triggered an intense discussion in the author group and we arrived at the following conclusions:
The suggestions for modification are as follows:
Comment 1: It is recommended to include the current domestic and international research status of distributed multi-sensor systems for robotic total stations and to conduct comparative research analysis.
Answer 1: After considering the feedback from Reviewer #3, we put the focus of the paper on the “UAS trajectory estimation” within decentralized multi-sensor systems based on RTS. Thus, we would see your recommendation in this very specific focus.
We added a tabular comparison in Section 2.3. Kinematic measurements of UAS with RTS addressing existing work that deals exactly with our topic “RTS for estimating UAS trajectories”, while also critically investigating the measurements. We feel the large number of publications using RTS but not investigating how good the measured trajectories actually are, give no added value in our article. Thus, we do not include publications that simply use RTS for the trajectory estimation of UAS.
Unfortunately, we do not agree with the separation in domestic and international research, as we focus on the state of the research in general.
Comment 2: It is recommended to make certain adjustments to the structure of the article. Currently, a large portion of the article is focused on describing RTS technology. Adding comparative analyses of related technologies could highlight the advantages of this technology.
Answer 2: We agree that the motivation for using RTS technology was a bit short. Thus, we extended the corresponding part in the introduction to properly motivate why other techniques can not be used for highly-accurate UAS trajectory estimation.
However, we feel including a thorough review of alternative trajectory estimation technologies (GNSS, SLAM, Visual Odometry etc.) is beyond the scope of our paper.
Comment 3: It is recommended to add some of the latest references in this field, which can be used for comparison with RTS technology.
Answer 3: Our article reviews the current state-of-the-art of decentralized multi-sensor systems based on RTS in the context of UAS trajectory estimation. We feel that we included the most relevant and latest reference in our work. We hope that you agree that a detailed comparison between e.g. RTS, GNSS, SLAM, Visual Odometry etc. is beyond the scope of our contribution.
Thank you,
The Authors
Reviewer 3 Report
Comments and Suggestions for Authors
The paper “A review on decentralized multi-sensor systems based on robotic total stations” is devoted to analysis of total station applications for trajectory estimations of unmanned aerial systems. The paper is well-written and clear, its structure is adequate. There are several comments, which could slightly improve the text.
The review is mainly about trajectory estimation of UAS, it would be better if this was reflected in the paper title. Something like “A review on UAS trajectory estimation using decentralized multi-sensor systems based on robotic total stations”, would probably be more informative.
The concept of “decentralized multi-sensor system is not clear”. In lines 45-47 the authors define «centralized multi-sensor systems typically realize time synchronization through wired connections, whereas decentralized systems require a wireless approach, which can be realized using the satellite-based pulse per second signal». So, the main distinction between these types is the way of time synchronization?
Please, provide the dimensions of the amplitudes and ranges in lines 83-85.
Make sure that all abbreviations are explained after its first appearance in the text. For example, EDM is mentioned in line 66, though its explanation is in line 252.
Author Response
Dear Reviewer,
Comment 1: The paper “A review on decentralized multi-sensor systems based on robotic total stations” is devoted to analysis of total station applications for trajectory estimations of unmanned aerial systems. The paper is well-written and clear, its structure is adequate. There are several comments, which could slightly improve the text.
Answer 1: Thank you for the very detailed and kind feedback. We feel your comments are very adequate and helped a lot to clarify our work.
Comment 2: The review is mainly about trajectory estimation of UAS, it would be better if this was reflected in the paper title. Something like “A review on UAS trajectory estimation using decentralized multi-sensor systems based on robotic total stations”, would probably be more informative.
Answer 2: It is correct that our article almost exclusively addresses the trajectory estimation and this fact should be represented in the title. We changed the title accordingly.
Comment 3: The concept of “decentralized multi-sensor system is not clear”. In lines 45-47 the authors define «centralized multi-sensor systems typically realize time synchronization through wired connections, whereas decentralized systems require a wireless approach, which can be realized using the satellite-based pulse per second signal». So, the main distinction between these types is the way of time synchronization?
Answer 3: No, the main distinction is the spatial distribution of the sensors. This consequently affects the possible time synchronization between the sensors. We added a clear distinction between decentralized and centralized systems and use these terms based on the spatial distribution of the sensors.
Comment 4: Please, provide the dimensions of the amplitudes and ranges in lines 83-85.
Answer 4: The refraction coefficient is a unit-less value and thus the amplitude and ranges are as well. To make this easier to understand for the reader, we added an explanation on the refraction coefficient in the respective part.
Comment 5: Make sure that all abbreviations are explained after its first appearance in the text. For example, EDM is mentioned in line 66, though its explanation is in line 252.
Answer 5: The abbreviation of EDM was unfortunately swapped with another occurrence. Thus, we failed to change its explanation accordingly. We checked the paper again and are now sure that all abbreviations are explained at their first occurrence.
Thank you,
The Authors
Round 2
Reviewer 1 Report
Comments and Suggestions for Authors
1. The title of the paper has been changed to a review. Since it is a review paper, the writing method of a review paper should also be changed accordingly.
2. The abstract section does not follow the writing method of an academic paper's abstract. The first sentence should present the core research content and key points, thereby introducing the key points of this paper.
3. The abstract of a review paper should have the framework of a review paper, outlining what content is reviewed, what conclusions are drawn, and which methods are helpful for the author's main innovative points.
4. The abstract should be concise and to the point. Sentences like the above should not appear in the abstract.
5. In the introduction and literature review sections, since the theme has been changed to a review, the time and key research points should be highlighted as the clues for organization. Moreover, the paper should not only describe the research of others but also provide the author's analysis and discussion.
6. In a review article, after analyzing and summarizing others' papers, several major future research directions and key technical points should be generated.
7. The reading volume of review articles is only 78, which is obviously insufficient.
Author Response
Dear Reviewer 1,
Thank you again for your feedback. Please find our detailed answers below:
Comment 1: The title of the paper has been changed to a review. Since it is a review paper, the writing method of a review paper should also be changed accordingly.
Answer 1: The title was changed from “A review on decentralized multi-sensor systems based on robotic total stations” to “A review on UAS trajectory estimation using decentralized multi-sensor systems based on robotic total stations”, but the keyword review was always included. Thus, neither the scope nor the type of the paper was changed. We therefore belief that the writing method does not need further adaption.
Comment 2: The abstract section does not follow the writing method of an academic paper's abstract. The first sentence should present the core research content and key points, thereby introducing the key points of this paper.
Answer 2: Thank you for pointing this out, we adjusted the abstract accordingly to start with the appropriate sentence.
Comment 3: The abstract of a review paper should have the framework of a review paper, outlining what content is reviewed, what conclusions are drawn, and which methods are helpful for the author's main innovative points.
Answer 3: Although we would like to follow your recommendation, we do not see a possibility to provide a complete overview of (i) the reviewed topics, (ii) the drawn conclusion and (iii) a judgement on the helpfulness of the reviewed content with a 200-word limit. In its current form, the abstract is already above the limit and gives an overview of the general topic as well as pointing out the conclusions.
Comment 4: The abstract should be concise and to the point. Sentences like the above should not appear in the abstract.
Answer 4: May we please ask you to state which sentence you are referring to?
Comment 5: In the introduction and literature review sections, since the theme has been changed to a review, the time and key research points should be highlighted as the clues for organization. Moreover, the paper should not only describe the research of others but also provide the author's analysis and discussion.
Answer 5: The theme has always been a review. However, we organize our article according to thematic topics. The time aspect is only relevant for hardware-dependent sources. We introduce the structure of our article at the end of the Introduction. Also we provide an analysis and contextualisation of the reviewed articles in the Conclusion section.
Comment 6: In a review article, after analyzing and summarizing others' papers, several major future research directions and key technical points should be generated.
Answer 6: We synthesize the state-of-the-art in the Conclusion section and, based on that, derive and state our major future research directions.
Comment 7: The reading volume of review articles is only 78, which is obviously insufficient.
Answer 7: The number of references was 84. We agree that this is a relatively low number for a review paper, but it simply reflects the sparsity of research on the topic of RTS-based multi-sensor systems.
Reviewer 2 Report
Comments and Suggestions for Authors
All the reviewers' concerns have been adequately addressed in the revised manuscript, making it suitable for acceptance.
Author Response
Dear Reviewer,
thank you for helping us to improve and accepting our (revised) manuscript.
All the best,
The authors